# “NETs and EETs, a Whole Web of Mess”

**DOI:** 10.3390/microorganisms8121925

**Published:** 2020-12-04

**Authors:** Tyler L. Williams, Balázs Rada, Eshaan Tandon, Monica C. Gestal

**Affiliations:** 1Department of Microbiology and Immunology, Louisiana State University (LSU), Health Science Center, Shreveport, LA 71103, USA; twil63@lsuhsc.edu (T.L.W.); eshaantandon3@gmail.com (E.T.); 2Department of Infectious Diseases, University of Georgia, Athens, GA 30302, USA; radab@uga.edu

**Keywords:** neutrophils, eosinophils, neutrophil traps, eosinophil traps, bacteria immunomodulation, immunology

## Abstract

Neutrophils and eosinophils are granulocytes that have very distinct functions. Neutrophils are first responders to external threats, and they use different mechanisms to control pathogens. Phagocytosis, reactive oxygen species, and neutrophil extracellular traps (NETs) are some of the mechanisms that neutrophils utilize to fight pathogens. Although there is some controversy as to whether NETs are in fact beneficial or detrimental to the host, it mainly depends on the biological context. NETs can contribute to disease pathogenesis in certain types of diseases, while they are also undeniably critical components of the innate immune response. On the contrary, the role of eosinophils during host immune responses remains to be better elucidated. Eosinophils play an important role during helminthic infections and allergic responses. Eosinophils can function as effector cells in viral respiratory infections, gut bacterial infections, and as modulators of immune responses by driving the balance between Th1 and Th2 responses. In particular, eosinophils have biological activities that appear to be quite similar to those of neutrophils. Both possess bactericidal activity, can activate proinflammatory responses, can modulate adaptive immune responses, can form extracellular traps, and can be beneficial or detrimental to the host according to the underlying pathology. In this review we compare these two cell types with a focus on highlighting their numerous similarities related to extracellular traps.

## 1. Introduction 

The immune system is a complex myriad of signals that orchestrates the timing, length, and quantity of the immune response [1]. A classic immune response consists of very well-coordinated steps and begins with innate immunity providing the first level of protection. The innate immune response includes phagocytic cells and physical barriers at mucosal sites [2,3,4,5,6]. After several steps, the innate response is followed by adaptive immunity that will be responsible for the ultimate, long-term clearance of the infection. The successful generation of protective immunity is dictated by a finely-tuned slew of immune signals beginning with those produced by innate cells [3,6,7,8,9]. Nevertheless, while host immunity fights to orchestrate the battle against the external aggression, microbes utilize means to disrupt this precise and well-adjusted communication. The manipulation and disruption of the immune homeostasis can lead to overreactive proinflammatory responses that can cause tissue damage, disease, and even death of the patient, as is currently happening during the COVID-19 pandemic.

Polymorphonuclear granulocytes are white blood cells characterized by harboring numerous cytoplasmic granules and segmented nuclei. Granulocytes are a very important part of the first line of the immune response against infections or external threats [10]. Beyond their classical role, their outstanding and recently-recognized versatility allows them to work as antigen presenting cells [10,11], modulators of adaptive responses [12,13,14,15], and overall shapers of the immune memory [10,16]. There are three major classes of granulocytes—neutrophils, eosinophils, and basophils, although mast cells can also be included. Neutrophils represent the most numerous and best-studied class [10].

In this review, we focus on a special function of these cells, which is the formation of extracellular traps (ETs). We first highlight neutrophils as we talk about their role as immune cells, the traps they form, and the functionality of these traps. This will be followed by a comparison between neutrophils and eosinophils as well as the traps they both form, and we outline similarities and differences between them. We will also dedicate a small section to exploring how pathogens can escape or evade neutrophils and their traps, as well as the recent discovery of bacterial modulation of eosinophils [17]. We give examples of some of the pathogens executing these defense mechanisms as well as some of the diseases that involve them (Figure 1). We conclude by discussing future directions focusing on the application of current knowledge for the development of therapies as well as guiding future research directions regarding these two cell types.

## 2. Neutrophils and Neutrophil Extracellular Traps

Neutrophils are a very abundant cell type in humans (55–70% of all peripheral white blood cells). They respond very quickly to infectious agents by trapping the microbes and releasing antimicrobial molecules with the purpose of killing them [18,19,20,21,22]. They are formed and differentiated in the bone marrow before being released into the circulatory system. Neutrophils contain many granules that are classified into three main groups, (1) azurophilic granules, that contain myeloperoxidase (MPO), bactericidal/permeability-increasing protein (BPI), defensins, neutrophil elastase (NE), and cathepsin G; (2) specific granules that encompass alkaline phosphatase, lysozyme, nicotinamide adenine dinucleotide phosphate oxidase (NADPH oxidase), collagenase, lactoferrin, histamine, and cathelicidin; and, (3) tertiary granules that have cathepsin, gelatinase, and collagenase. There is a fourth type of granule called a secretory vesicle. These are not real granules, but vesicles derived from the plasma membrane. These vesicles are important in early events of polymorphonuclear activation and largely responsible for the many versatile functions of neutrophils [23].

Although neutrophils are short-lived, they are very motile and can travel through tight junctions between cells allowing them to reach a multitude of different body compartments. An interesting aspect of these cells is their unique ability to form neutrophil extracellular traps (NETs) [24,25,26,27] allowing them to catch pathogens in the extracellular space in order to control the spread of the infection. NETs can trap and kill microbes which significantly decreases the microbial burden [28,29,30] aiding the immune system in the clearance of the infection and in the individual’s overall survival. 

### 2.1. Neutrophils in Infection

Neutrophil-mediated microbial clearance is required in infections caused not only by extracellular [31,32] but also intracellular pathogens such as *Listeria monocytogenes* [33]. Neutrophils’ classical role during the early innate immune response involves bacterial phagocytosis and killing facilitated by the generation of reactive oxygen species (ROS), NET formation, and production of proinflammatory cytokines [19,34,35]. However, current research has demonstrated that neutrophils can also acquire and perform other important immune functions including antigen presentation [11] and modulation of adaptive immunity [12].

Neutrophils are essential during the signaling cascade required to activate an efficient inflammatory response. They produce a plethora of chemokines and cytokines [36,37] highlighting the diversity of neutrophils in the modulation and overall conservancy of the immune homeostasis. While their job as first line of defense against pathogens is unquestionable, the formation of NETs remains a controversial function of neutrophils due to their critical role during bacterial clearance, while simultaneously associated with deleterious auto-inflammatory and autoimmune diseases.

### 2.2. NETs 

NETs are made of extracellular fibers whose backbone consists of DNA [38,39]. NETs are complex structures formed not only by chromosomal and mitochondrial DNA, but also DNA-associated histones and granule proteins including neutrophil elastase (NE), cathepsin G, and MPO [25,40]. NETs can bind pathogens that adhere to the DNA through the bacterial lipopolysaccharides, for instance [40,41]. This mechanism of defense is inducible by Gram-positive and Gram-negative pathogens [28,29,30,42,43], such as *Staphylococcus aureus* [44,45,46], *Escherichia coli* [47,48,49], or *Pseudomonas aeruginosa* [41,50,51,52,53,54]. 

The NET formation process has been referred to as NETosis [55]. It was thought that neutrophils have to die to form NETs and the actual cell death is called NETosis. However, not all neutrophils have to die in order to release the mitochondrial DNA. Yousefi et al. [56] demonstrated that pure mitochondrial DNA can form NETs while neutrophils remain viable. This means that neutrophils have a cytolytic and non-cytolytic mechanism for NETosis [56,57]. However, the molecular mechanism by which NETosis is induced and mediated is still not fully elucidated, albeit many pathways have already been implicated. NET formation was first described to require the release of reactive oxygen species (ROS) produced by the NADPH oxidase enzyme complex [28,55]. The activation of the NADPH oxidase is one of the main effector responses of neutrophils to external pathogens and it is also critical for NET formation due to its ability to activate intracellular granular proteases [58,59,60]. The localization (intra- vs. extracellular) of ROS production in neutrophils has been proposed to drive effector responses of neutrophils including NET extrusion to microbes of different sizes [61]. While several stimuli trigger NETs in an NADPH oxidase- and ROS-dependent fashion, numerous reports have emerged proposing an NADPH oxidase-independent mechanism of NET formation in response to specific stimuli including microcrystals [27,62,63,64]. 

Protein arginine deiminase 4 (PAD4), an enzyme performing protein citrullination as a post-translational modification, has been shown to be critical for NET formation [65,66]. PAD4 is highly expressed in neutrophils and localizes to the cytosol in resting cells [67,68]. Upon neutrophil stimulation, PAD4 translocates to the nucleus to promote histone citrullination that mediates chromatin disassembly [65,69,70]. This is followed by the disruption of the cytoskeleton, endomembranes, and the nuclear envelope [71]. Different protein kinase C isoforms have been implicated in the mediation of PAD4- and NADPH oxidase-mediated NET formation [72]. But regardless of the path utilized to initiate NET formation, once the process is activated, NE, MPO, and other neutrophil granule proteins are released and decorate the DNA web [73]. 

NETosis was proposed to occur by two different pathways: suicidal NETosis and vital NETosis [26,40,55]. Suicidal NETosis refers to NET formation due to the release of DNA following the death of neutrophils [74]. Vital NETosis is activated by pathogens, bacterial lipopolysaccharide, TLR4-activated platelets, or complement proteins. It is a rapid process that does not result in immediate neutrophil death, however, the cell loses its DNA. More intriguingly, after vital NETosis neutrophils are still capable of phagocytosing bacteria [75] suggesting that albeit not containing nucleic acid they are still able to perform biological functions for some time. This is in line with classical observations of human neutrophil ‘ghosts’ or ‘cytoplasts’ that are isolated neutrophils whose nuclei were removed experimentally but still remained capable of migration, phagocytosis, and ROS production [76,77].

### 2.3. NETs and Infectious Diseases

NETs are highly conserved amongst different species in the animal kingdom, which features its cruciality. As with any other immune function, the ability to form NETs appears to be affected by age. Previous reports demonstrate that neonatal neutrophils are less capable of forming NETs than adult neutrophils indicating that the immune system needs to acquire a certain degree of maturity in order to establish proper NETs [78]. 

As previously mentioned, there is an extensive number of pathogens that trigger NET formation by diverse mechanisms including lipopolysaccharide and flagella [79,80]. NETs can have bacteriostatic and/or bactericidal effects [28]. Within NETs, bacteria cannot utilize the bacterial virulence factors to downregulate host immune response allowing for the immune system to better control the infection [28]. This undeniably powerful mechanism provides protection against bacterial infections [28,29,51,81,82], mycobacteria [40], fungi [83], viruses [84,85], and parasites [86,87,88] offering a basic universal method to combat microbes. Thus, this highlights the very significant role of NETs during clearance of multiple kinds of infections. 

There is, nonetheless, debate in regard to the role of NETs during several chronic diseases such as cystic fibrosis (CF). CF is a lung disease where there is a build-up of chronic inflammation and thick mucus in the respiratory airways as well as robust bacterial colonization. Many different kinds of bacteria are found in CF patients including *Haemophilus influenzae, Staphylococcus aureus*, *Pseudomonas aeruginosa*, and *Bordetella* spp. [89]. As neutrophils release DNA, NETs are formed, and they promote the patients’ mucus to become thicker resulting in the build-up of a niche with a high amount of glycans that promotes a great bacterial diversity. NETs increase sputum viscosity causing the patients’ breathing capacity to decrease, negatively impacting their health status. Many CF patients are subjected to DNase treatment to disrupt the NETs so that sputum can be liquified and mucus can be cleared [90]. 

Neutrophils and their NETs play a very important role in the defense against fungal infections as well. Fungal infections are mostly controlled by neutrophils and they eradicate fungi by oxidative burst, phagocytosis, or releasing NETs. *Candida albicans* is a fungus that can cause infections in the gastrointestinal tract, mouth, throat, and the skin. While NETs bind to this fungal pathogen, neutrophils also release the protein calprotectin, which is a metal ion chelator. Calprotectin performs its antifungal activity by depleting Zn^2+^ and Mn^2+^. These two ions are very significant for *Candida albicans* to thrive, and NETs allow for calprotectin to better reach all fungi leading, then, to the control of the infection and its dissemination [83].

NET induction in viral infections has not been well-studied but scientists are now learning more and more about how NETs help protect the host during these infections. Neutrophils are able to recognize different viral pathogens and trigger NET formation via Toll-like receptors (TLR) like HIV-1, which promotes NETs via TLR7 and TLR 8 [84]. However, viruses can also promote NETosis indirectly without using pattern recognition receptors (PRRs). Chemokines and cytokines like interleukin-8 (IL-8) and type 1 interferons are present in virus-infected cells triggering NETosis in recruited neutrophils. These NETs help clear viral infections by binding to the viral particles and immobilizing them by electrostatic attraction, causing the virus to mechanically stop spreading [91]. But it is worth noting that bacterial and viral infections manipulate the host system in a very conservative way. This indicates that a better understanding of the precise mechanisms that pathogens utilize to modulate NETs can result in targets for therapeutics that can be applied to a broad spectrum of diseases.

#### NETs and COVID-19

In December 2019, a cluster of pneumonia of unknown etiology was identified in Wuhan, China. Rapidly, this novel virus known as SARS-CoV-2 emerged and it has spread worldwide causing a great pandemic during 2020. Since this pandemic started, many efforts have tried to characterize the pathology and immune responses to this novel coronavirus and a particular feature is the hyperreactive uncontrolled immune response that leads to the detriment of the patient’s health [92,93,94]. 

Increased neutrophil counts in the peripheral blood have been associated with a worsening of the patient’s prognosis. In fact, the neutrophil-to-lymphocyte ratio (NLR) has been used as a marker of severe disease [95,96,97]. In COVID-19 patients, reactive oxygen species (ROS) produced by neutrophils are augmented which can also contribute to organ failure [98]. 

Neutrophil degranulation and NET markers have been associated with severe illness during COVID-19 [99]. In fact, higher levels of NETs have been correlated with an increased death rate in COVID-19 patients, in which an overreactive neutrophilic response leads to a general multi-organ failure [97]. Neutrophils appear to be decorated with thrombocytes in severe cases of COVID-19, the number of NETs identified in the circulation is significantly increased in these patients [100,101,102] and this correlates with the evidence of increased platelet factor 4 detected in these patients [101]. This increased NET formation has also been associated with COVID-19-related acute respiratory distress syndrome and is a potential biomarker for disease severity [101]. SARS-CoV-2 has been shown to trigger NETs directly in neutrophils [103]. Because of all of this, neutrophils have emerged as potential, clinically relevant therapeutic targets in COVID-19 [104].

### 2.4. NETs and Inflammatory Diseases

While the benefit of NETs is to clear pathogens, the unchecked deposition of NETs has been associated with deleterious effects to health [105]. NETs have been related to several acute diseases such as acute respiratory distress syndrome [105], hepatic damage during bloodstream infection [106,107], as well as chronic diseases including cystic fibrosis lung disease [26,50,51,52] and chronic obstructive pulmonary disease (COPD) [108,109,110,111]. In CF patients, NETs associate, as previously mentioned, with an obstruction of the lung and decrease of lung function [80,112,113]. However, NETs are also involved in autoimmune diseases, cancer, and even fertility [114]. And excitingly, NETs and EETs, as discussed later, have also been associated with gout and its pathogenesis induced by the formation of monosodium urate microcrystals [27,62,115]. 

NETs play a very significant role in many diseases and health conditions including asthma and allergies. Unfortunately, NETs have a negative impact in asthmatic patients. In some cases when there is a disordered regulation in the formation of NETs, it can actually promote exacerbation of severe asthma and COPD112. 

Furthermore, it has been proven that NETs have been found in the sputum of severe asthmatic patients. A previous study was completed to compare the sputum of severe asthmatic patients to healthy control subjects by measuring extracellular DNA (eDNA). This study resulted in the knowledge that a higher eDNA concentration was found in severe asthmatic patients and that NETs mediate inflammasome activation as well as cell injury. High eDNA in the sputum of severe asthmatic patients correlated with their levels of NETs that were also cytotoxic to airway epithelial cells [116].

NETs have also been shown to have a role in thrombosis. Thrombosis is the pathological deviation of hemostasis which involves the mechanisms that stops bleeding after blood vessels are injured [117]. In other words, thrombosis is the creation of a blood clot in a blood vessel which stops leakage of blood into surrounding tissues but can also result in the restriction of blood flow. It has been shown that thrombosis depends on platelet aggregation and adhesion, and NETs that are perfused within the blood can accelerate platelet aggregation [118,119]. Deep vein thrombosis is a serious issue because it develops pulmonary embolisms, referred to as venous thromboembolism [120], and there are occlusions of the pulmonary artery due to the thrombus breaking apart and traveling to the lungs [117]. Studies regarding deep vein thrombosis have shown that NETs stimulate thrombus formation through the entrapment of red blood cells which also promotes fibrin deposition [118]. The results of these studies also indicated that NETs in sepsis have the same function [118]. Other studies have illustrated the role of NETs in acute myocardial infarction, revealing that NETs at the site of plaque rupture were enveloped by tissue factor which is a significant *in vivo* initiator of coagulation [121]. Overall, NETs contribute to several pathologies via their prothrombotic effect.

### 2.5. Bacterial Mechanisms to Escape NETs

As NETs remained well-conserved across evolution, and due to the great selective pressure that this possesses on infectious agents, bacteria have evolved mechanisms to escape or manipulate NET mediated clearance [122,123,124,125,126,127,128]. Since this topic has been extensively covered in several reviews [50,84,128,129,130], only a few highlights are mentioned here. *Bordetella pertussis* harbors many immunomodulators that control several aspects of the host immune response [1,131,132] including NETs. *B. pertussis* suppresses the generation of ROS inhibiting NET formation. Also, adenylate cyclase toxin (ACT) of *Bordetella* spp. contributes to the suppression of ROS [126]. Similarly, *Pseudomonas aeruginosa* possesses means to trigger NET formation. Swimming motility mediated by the bacterial flagellum has been shown to be an essential factor in triggering NETs in neutrophils [80]. Interestingly, it has been demonstrated that NET formation is also linked to quorum sensing (QS). Quorum sensing is the mechanism by which bacteria communicate with each other via diffusible molecules [133,134,135]. LasR, a regulator of QS that controls expression of many bacterial virulence factors, provides an exciting target to explore the relationship between NETs and QS. Previous results demonstrated that LasR-deficient mutants presented a defect in NET formation. This implies that there is a two-way signaling mechanism involved in the formation and maturation of NETs [136]. On one side, the bacteria might be sensing the host inflammatory response and responding by increasing the expression of factors that allow the blockage of host immunity. In parallel, the host immunity is able to detect cues of the bacterial threat and counteract accordingly by orchestrating signals that will allow for the required response that leads to clearance. A better understanding of this universal language will enable us to design immunomodulators that can be used not only for treatments but also vaccine design.

These are only two examples that highlight the subtleties in the regulation of NETs in immune responses. The fact that SARS-CoV-2 is also manipulating NETs suggests that disturbing this mechanism of defense provides the pathogen with a very powerful tool to cause infection. 

## 3. Eosinophils and Eosinophil Extracellular Traps

### 3.1. Eosinophil Biology

Eosinophils are terminally differentiated granulocytes that develop in the bone marrow from granulocytic/myeloid progenitors in mice and common myeloid progenitors in humans [137]. They become eosinophil progenitor cells (EoP) and differentiate into mature eosinophils prior to exiting the bone marrow [138]. Eosinophils rely on cytokines like IL-3, IL-5, and Granulocyte-macrophage colony-stimulating factor (GM-CSF) for hematopoiesis, survival, and enhanced activation by other stimuli. Homeostatic migration of eosinophils occurs through eotaxin-1 (CCL11), while eotaxins -2 (CCL24) and -3 (CCL26: human only) are the primary mediators of their migration into tissues in disease states. They are found in low numbers in the circulation (1–3% of all white blood cells) and can be resident cells in tissues such as the gastrointestinal tract, lung, thymus, adipose tissue, and endometrial tissue, as well as in secondary lymphoid tissues [139,140,141]. 

Eosinophils contain eosinophil-specific (secondary) granules [129] that are released by several mechanisms into the extracellular milieu. Within these secondary granules are proteins that are conserved across species like major basic protein (MBP) and eosinophil peroxidase (EPX). There are also granule matrix proteins like eosinophil cationic protein (ECP) and eosinophil-derived neurotoxin (EDN) that are divergent in mice forming a family of murine ribonucleases (mEARs) [142]. Importantly, eosinophil granules also contain immune signals such as chemokines, growth factors, and cytokines [143,144,145] within their granules that can be actively and specifically released without requiring previous RNA transcription, leading to a very rapid response. Granules and granule contents are released via several mechanisms in response to a variety of stimuli. This includes, for example, specific release of mediators through piecemeal degranulation or release of entire granules resulting in cell-free granules that deposit in tissue, or through lysis of the cell. 

The classic role of the eosinophil is described as that of a cell that kills and damages other cells, including pathogens. For example, eosinophils contain antimicrobial peptides (AMPs) and generate reactive ROS that kill pathogens. Although eosinophils are known by these innate features that lead to tissue damage or cell death, eosinophils are increasingly being defined as cells with the potential to modulate immune responses as well as remodel and repair tissue as described by Lee and colleagues [146].

For example, eosinophils have been identified as cells that respond to innate cytokines such as IL-33, an IL-1 family cytokine [147], to induce type 2 cytokine production, macrophage polarization, and dendritic cell activation [148,149]. Moreover, eosinophils may modulate the polarization and activation of T cells through direct and indirect mechanisms. Tissue remodeling and repair by eosinophils has been shown to be somewhat dependent on the extracellular matrix components that, in turn, may activate or suppress eosinophils [150]. Finally, eosinophils have the capacity to resolve inflammation, whether directly, by the release of resolvins [151], or through activation of phagocytic macrophages [152]. The roles of eosinophils in infectious responses as well as non-allergic diseases have enhanced our understanding of these cells as complex participants in human health [137,139,153,154]. 

### 3.2. EETs and EETosis

The discovery of eosinophil extracellular traps (EETs) is a novel exciting area of investigation that promises to unravel important functions for these cells. EETs, similar to NETs, have many advantages when it comes to infectious diseases, although this comes with a price [73].

Eosinophil traps have been shown to occur in humans as well as other animal species [155,156]. The original hypothesis of how these eosinophil extracellular traps are formed was demonstrated by Yousefi and colleagues proposing that similarly to neutrophils, eosinophils can eject their mitochondrial DNA and granule content while still remaining viable cells [157]. A second mechanism was identified by Ukei and colleagues whereby lytic cells released nuclear DNA that contained histones and cell-free secondary granules. When the cells are presented with immobilized immunoglobulins (IgA and IgG), calcium ionophore, platelet activating factor, or phorbol myristate acetate, cellular death and degranulation are promoted [141,142] and EETosis takes place [143]. It is possible, as well, that NADPH oxidase-dependent mechanisms, similar to NETosis [158], are needed for EETosis. Recent reports suggest several mechanisms may give rise to either lytic or viable DNA release from eosinophils, all of which require further study. However, there is evidence that extracellular DNA trap formation for viable and lytic EETosis happens *in vivo* [148].

Monosodium urate crystals (MSU) also induce trap formation in neutrophils and eosinophils [159,160]. MSUs have been associated with the enhancement of the autoinflammatory disease gout [161], partially due to their ability to trigger NETs and EETs. Interestingly, this poses the question as to whether there is a feedback regulatory mechanism between both cell types and their traps.

### 3.3. Eosinophils and Their Traps in Infectious Diseases 

Eosinophils have been shown to lead to the clearance of specific parasitic infections [162], bacterial infections, and viral infections [163,164,165,166,167,168]. Even though the role of eosinophils during infection has only started to be delineated, some evidence of their important involvement in immunity to bacterial infections has been noted mostly in two models of infection. During *Staphylococcus aureus* septicemia, the classical proinflammatory response involves production of type 1 and type 17 T helper cells [102]. Additionally, preliminary data have revealed that the cytokine secretion by group 2 innate lymphoid cells (ILC2) combined with eosinophil function can skew the response towards a type 2 response, and this appears to be a better defense mechanism against *S. aureus* infection [169]. Septicemia survivors present higher numbers of Th2 cells [169] indicating that eosinophils mediate the generation of efficient immunity during *S. aureus* systemic infections—by a still unknown mechanism.

An alternative model to study the role of eosinophils in inflammatory responses has also been revealed during *Helicobacter pylori* infection. *H. pylori* is a Gram-negative bacterium that is generally found in the digestive tract. During *H. pylori* infection, eosinophils promote immune responses by suppressing Th1 immune responses allowing for a robust Th2 response that aids clearance [170]. 

An important role of eosinophils and EETs is highlighted during fungal infections. Fungi are not only pathogens, but they can also be major allergens [171] triggering allergic responses classically associated with eosinophils. ABPA (allergic bronchopulmonary aspergillosis) is involved in an allergic response to *Aspergillus fumigatus,* a fungus that is commonly found in patients with cystic fibrosis and asthma [172]. ABPA can cause such a severe eosinophil-dependent asthmatic response that can lead to a significant decrease in lung function [171]. Following direct interaction with *A. fumigatus*, eosinophils go through EETosis contributing to the thickening of the mucus and eventually causing mucus plugging [173]. Mucus plugging is associated with recurrent relapse, and it is thought to be the cause of severe inflammation in the airways [173]. Furthermore, the induction of the EETs by this particular fungus culminates in an ROS-independent process [174]. Altogether, this indicates that eosinophils likely play an important role in the immune responses to *A. fumigatus* and their dysregulation may lead to increased morbidity associated with the primary infection [92]. 

Paracoccidioidomycosis (PCM) is caused by the fungus known as *Paracoccidioides brasiliensis* and eosinophils are also involved in the pathology of PCM disease [175,176,177]. Patients suffering from PCM have declining T cell immunity and an increasing number of blood eosinophils, which, combined, induces high levels of inflammatory response in the very first phase of the infection [178]. Moreover, animal experiments have revealed that increased levels of IL-5 result in the blocking of eosinophil maturation increasing the susceptibility to PCM. The association between eosinophils and disease severity implies that immunomodulators of eosinophils might be a good therapeutical candidate for these fungal diseases.

An example also related to *S. aureus* is eosinophilic chronic rhinosinusitis (CRS). Eosinophilic CRS is a refractory disease characterized by the presence of thick mucus, nasal polyps, and congestion. Eosinophil infiltration in these patients is often associated with secondary infections that triggers proinflammatory responses [179]. EETs are formed at the sites of airway damage and they protect CRS patients from external aggressions like infections by entrapping bacteria such as *S. aureus* [180]. 

Eosinophils also play a role in many viral infections [181,182,183]. A study was completed looking at how eosinophils respond to influenza A virus infection [184]. This study revealed that eosinophils could act directly against influenza A, but they also mediated CD8+ T cell responses against the virus [182]. Similarly, respiratory syncytial virus (RSV) also induces EET formation in vitro in a concentration-dependent manner [181]. Interestingly, RSV is frequently related to asthma [185] exacerbations, further implicating a role for eosinophils in these infections. RSV induces EETs in the bronchoalveolar lavage fluid in mice with asthma which is suggested to promote airway obstruction, mucus plugs, and inflammation in respiratory airways [181]. 

Altogether, this indicates that eosinophils likely play significant roles during infections, however, the recent attention focused on these cells will soon enlighten more of the interesting biology of eosinophils and EETs and their roles in the immune responses to infectious diseases. The function of EETs in pathogen clearance and the precise mechanisms that trigger EET formation and modulation still remain not fully understood. 

#### Eosinophils and COVID-19 Infection

Persistent eosinopenia has been directly correlated with severe COVID-19 with a positive predictive value of over 72% [186,187,188,189,190,191,192,193,194,195,196], and eosinophil counts on peripheral blood have been proposed as biomarkers [187,197]. Eosinophil counts appear to be very low, with a minimum at day 4 [197] when patients arrive to the hospital and the numbers slowly increase in those patients that recover, while continuing at minimum levels for those with severe manifestations of the disease [196]. COVID-19 patients have low blood eosinophils, lung biopsies are characterized by the lack of eosinophils and massive macrophage infiltration. However, eosinopenia has also been thought of as a sign or consequence of host exhaustion from clearing the COVID-19 virus [198]. Dr. Iwasaki and colleagues characterized the immune responses to COVID-19 in a cohort of patients with different stages of severity where correlations between elevated type 2 responses in severe COVID-19 infections appear [199]. Patients with severe disease showed increases in eosinophils and levels of eotaxin-2, IL-5, and IL-13 compared to those with moderate disease. The apparent association with eosinophils and other signatures that are generally associated with Th2 responses, was unexpected [182].

Two eosinophilic proteins, eosinophilic cationic protein and eosinophil-derived neurotoxin (EDN), neutralize SARS-CoV-2 [200] and strategize to avoid clearance and increase persistence while keeping low levels of eosinophils [201]. In fact, elevated levels of EDN in serum have been observed in children with persistent wheeze-associated illness. And interestingly, IL-5 and IgE have also appeared to be augmented in the worse cases of COVID-19 [199], suggesting that eosinophil signatures could be considered as indicators of patient prognosis. 

These, so far, are preliminary studies, and more investigations need to be done to evaluate the role of eosinophils during COVID-19 infections. Is the virus manipulating host response and promoting eosinopenia, are eosinophils forming traps, what are the consequences of this decrease in the numbers of eosinophils during early stages of infection? These and many other questions are still to be answer.

### 3.4. Bacterial Mechanisms to Escape EETs

The fact that EETs are a host mechanism to fight a variety of infectious diseases suggests that it is most likely that pathogens have evolved means to evade, manipulate, or escape EETs. The compelling evidence that EETs are involved in infectious pathologies such as septicemia or fungal and viral respiratory diseases leads to the question of the implication of these traps in other bacterial infections. A recent study reveals how well-adapted pathogens can efficiently block eosinophilic influx in the lungs, causing a defect in adaptive responses associated with an increased pathogen persistence [17]. The *Bordetella bronchiseptica* sigma factor, *btrS*, regulates many immunomodulators associated with suppressing different aspects of the immune response [202]. *B. bronchiseptica btrS*-null trigger a prompt and robust adaptive response [131]. Some insight into the possible mechanisms associated with this increase in adaptive responses points two different cells, macrophages and eosinophils. Transcriptomic data of macrophages challenged with this mutant indicate some Th2-related signatures [131]. Further studies in eosinophil-deficient mice, revealed that while the wildtype mice clear the infection in 14 days, mice lacking eosinophils present a long-term infections characteristic of the wildtype bacteria [17]. These results suggest that wildtype bacteria somehow block eosinophil influx in the lungs to prevent the generation of prompt and robust adaptive immunity and to promote long term bacterial persistence. 

These findings open a novel perspective and understanding of bacterial ability to manipulate host immune responses using very sophisticated mechanisms. This suggests that we might need to reconsider the classical view that we have of some immune cells.

### 3.5. EETs and Inflammatory Diseases

While inflammatory diseases caused by NETs are very diverse, eosinophils and EETs have been mostly associated with allergic [203] and non-allergic respiratory diseases. EETs can negatively affect patients’ health due to the increased production of mucus secretions [155]. Unfortunately, dysregulations of eosinophilic functions can lead to pathological disorders, such as atopic dermatitis, rhinitis, asthma, and other inflammation-associated medical conditions [137]. 

Allergic asthma [178] is generally a very complex disease characterized by reversible bronchial hyperresponsiveness often associated with airway eosinophilia. In particular, severe eosinophilic asthma, which is often refractory to inhaled corticosteroids, has been found to present neutrophil and eosinophil traps that may play a role in disease pathologies as well as serve as biomarkers [204,205]. EETs contribute to asthmatic patients having respiratory problems likely due to the EETs promoting mucus plugging in the airways and epithelial damage [206]. Overall, these activities of eosinophils may contribute to asthmatic exacerbations [207]. One mechanism of EET activity in asthma may be through the functions of granule proteins and granules associated with the DNA NETs, resulting in tissue damage and cell-free granule signaling [205,208]. To further support these findings, when studying the role of EETs in the murine model, EETs injected into naïve wild-type mice induced epithelial activation that promoted group 2 innate lymphoid cell activities thus increasing type 2 cytokines IL-5 and IL-13 in the airways and lungs [206]. Overall, this study showed that EETs could potentially modulate type 2 immune responses in severe asthmatic patients with high airway and blood eosinophilia.

A further study indicated that indeed extracellular granules are found in tissues of asthmatic patients and this implies that eosinophils release cytotoxic cellular content that causes damage to the tissues. There are certain signaling pathways involved in triggering cytolysis, the bursting of a cell due to an osmotic imbalance, and the release of these free granules of eosinophils. Many of the granules are activated by interleukin-3 (IL-3). This study looked at IL-3-primed eosinophils seated on immunoglobulins like IgG. When eosinophils were exposed to IL-3, they degranulated early in the presence of IgG and the cytolysis of the IL-3-primed eosinophils was dependent on ROS production and involved eosinophil migration and adhesion [209].

Non-respiratory diseases including thrombosis and atherosclerotic plaque formation have also been associated with EETs [210,211]. Previous data have revealed that EETs could be used as targets for atherosclerosis treatment, for example using Siglec 8/F antibodies to block EET formation [210], providing a novel approach for a disease that is highly prevalent. In fact, in mice, SiglecF antibodies can reduce thrombus stability preventing arterial thrombosis [210]. In this study, the data suggested that eosinophil EETosis and MBP release promote platelet activation, thrombus formation, and production plaques in blood vessels. As EETs have been associated with thrombosis formation in atherosclerotic plaques it is not without reason to suspect EETs may be present in other diseases with thrombus formation, such as chronic urticaria [212] and other skin diseases [213]. Altogether these data suggest a novel mechanism of eosinophil-mediated damage to vasculature and tissue.

## 4. Future Directions 

The role of eosinophils during immune responses against pathogens has been under-studied, nevertheless, when looking carefully, neutrophils and eosinophils share many functions. In this review we only explored one of the many aspects of this cell type, furthering the demonstration that EETs (of eosinophils) are similar to NETs (of neutrophils). 

Interestingly, NETs and EETs possess many common mechanisms and functions. It could even be possible that NETs trigger the formation of EETs by inducing signals such as MSU crystals indicating that these two processes might complement each other and be interconnected. But the question remains as to whether, if the microbes are able to manipulate NET formation, are they also able to manipulate EET formation? How do pathogens disrupt immune signals orchestrating NETs and EETs and promoting inflammation in autoinflammatory disorders? If pathogens disrupt immune signals causing overreactive proinflammatory responses, can we manipulate them artificially as a treatment for some of the aforementioned diseases? Also, does viable and lytic EETosis and NETosis happen at the same time *in vivo* and in the same diseases?

Both EETs and NETs have advantageous and disadvantageous effects for the host. The great opportunity comes with the gain in knowledge of EETs, as it can provide not only a better understanding of the biology of eosinophils and their role during inflammatory responses, but it can generate targets for therapeutic agents of many infectious and eosinophilic diseases.

## Figures and Tables

**Figure 1 microorganisms-08-01925-f001:**
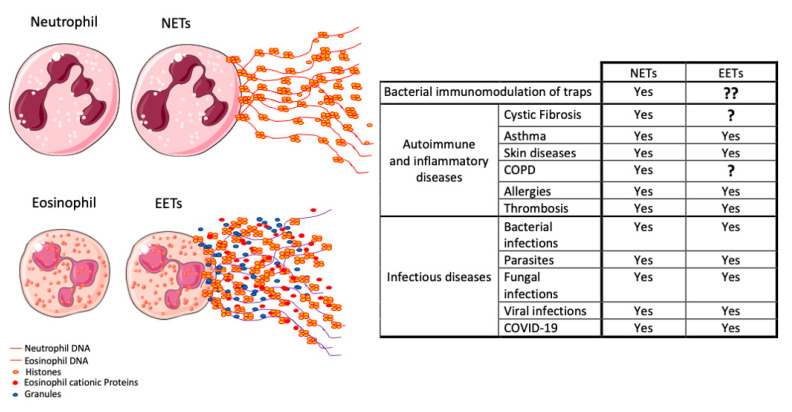
Involvement of NETs and EETs in diseases. The figure compares neutrophils and eosinophils as well as the extracellular traps they form. The right-hand side table summarizes autoimmune, autoinflammatory, and infectious diseases associated with the formation of NETs or EETs. The table also includes the ability of bacteria to modulate extracellular traps formation.

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
