# Peer review of "“NETs and EETs, a Whole Web of Mess”"

_microorganisms, 2020, doi:10.3390/microorganisms8121925_

Round 1
Reviewer 1 Report
I believe that a more extensive mention on the role of NETs and EETs in thrombosis would increase the impact of this review. NETs have been associated with thrombosis in multiple clinical scenarios such as deep venous thrombosis, acute myocardial infarction and sepsis.
References
- Fuchs TA, Brill A, Duerschmied D, Schatzberg D, Monestier M, Myers DD, Jr, et al. Extracellular DNA traps promote thrombosis. Proc Natl Acad Sci U S A(2010)
- von Brühl ML, Stark K, Steinhart A, Chandraratne S, Konrad I, Lorenz M, et al. Monocytes, neutrophils, and platelets cooperate to initiate and propagate venous thrombosis in mice in vivo. J Exp Med(2012)
- Massberg S, Grahl L, von Bruehl ML, Manukyan D, Pfeiler S, Goosmann C, et al. Reciprocal coupling of coagulation and innate immunity via neutrophil serine proteases. Nat Med(2010)
- Stakos D, Kambas K, Konstantinidis T. Expression of functional tissue factor by neutrophil extracellular traps in culprit artery of acute myocardial infarction. Eur Heart J (2015)
- Engelmann B, Massberg S. Thrombosis as an intravascular effector of innate immunity. Nat Rev Immunol(2013)
Also, since this is a review, a summarizing table (i.e. differences and similarities of NETs and EEs in mechanisms or in clinical situations involved) or an eye-catching figure (of both) would be helpful.
Sincerely
Author Response
Reviewer 1
I believe that a more extensive mention on the role of NETs and EETs in thrombosis would increase the impact of this review. NETs have been associated with thrombosis in multiple clinical scenarios such as deep venous thrombosis, acute myocardial infarction and sepsis.
We would like to thank the reviewer for the suggestion of expanding on the thrombosis, which is a highly relevant topic due to the high incidence of cardio-vascular diseases. We very much appreciate the suggestions regarding the literature which have been very helpful. Now we have a new paragraph dedicated to NETs and thrombosis in section 2.4.
We have also changed the graphical abstract which now includes a table with the pathologies associated with NETs and EETs.
References
Fuchs TA, Brill A, Duerschmied D, Schatzberg D, Monestier M, Myers DD, Jr, et al. Extracellular DNA traps promote thrombosis. Proc Natl Acad Sci U S A(2010)
von Brühl ML, Stark K, Steinhart A, Chandraratne S, Konrad I, Lorenz M, et al. Monocytes, neutrophils, and platelets cooperate to initiate and propagate venous thrombosis in mice in vivo. J Exp Med(2012)
Massberg S, Grahl L, von Bruehl ML, Manukyan D, Pfeiler S, Goosmann C, et al. Reciprocal coupling of coagulation and innate immunity via neutrophil serine proteases. Nat Med(2010)
Stakos D, Kambas K, Konstantinidis T. Expression of functional tissue factor by neutrophil extracellular traps in culprit artery of acute myocardial infarction. Eur Heart J (2015)
Engelmann B, Massberg S. Thrombosis as an intravascular effector of innate immunity. Nat Rev Immunol(2013)
Also, since this is a review, a summarizing table (i.e. differences and similarities of NETs and EEs in mechanisms or in clinical situations involved) or an eye-catching figure (of both) would be helpful.
We have changed the figure and now better summarizes the paper.
Sincerely
Reviewer 2 Report
General comments:
This review manuscript describes neutrophil and eosinophil extracellular traps (NETs and EETs), including the mechanisms of DNA release and trap formation, as well as the function of traps in microbial defense and presence in disease. The manuscript also to some extent covers neutrophils and eosinophils more generally, and compares the two cell types. The subject is overall very timely. However, clarifications are needed at various places in the manuscript and expanded discussion on some items would be interesting for the reader.
Major comments:
1. Does vital NETosis include mitochondrial DNA and a similar mechanism as non-cytolytic EETosis?
2. Do you believe that both viable and lytic EETosis happen in vivo or do you think vital mitochondrial DNA release may be a laboratory or experimental artefact? Do they both occur at the same time in the same diseases or are there differences?
3. E.g., on line 366: Please discuss how EETs can be targeted?
Minor comments:
4. Line 98: Should it be “MPO” instead of “MOP” here?
5. Line 140: What does the word “halt” mean and the entire sentence that includes the words “bacterial virulence factors are halt…”? Please rewrite.
6. Line 166: What is meant by “Neutrophils are able to recognize trigger NET formation…”? Recognize what?
7. Section 2.4 about NETs and inflammatory diseases including asthma: In this context, Lachowicz-Scroggins et al. 2019 (Am. J. Respir. Crit. Care Med. 199:1076-85) found NETs in sputum of a subset of patients with severe asthma and NETs were cytotoxic for airway epithelial cells.
8. Line 233: Is “secondary granule” correct current nomenclature for eosinophils? Please consult recent reviews on eosinophil granules by Weller, Spencer and Melo.
9. Section 3.3 regarding eosinophils presented with immunoglobulins (Igs), etc.: Pathways involved in triggering cytolysis of and release of free granules from eosinophils activated by interleukin (IL)-3 and plated on aggregated IgG were recently described (Clin. Exp. Allergy 2020, 50:198-212).
10. Lines 330-333: The relation between eosinophils and COVID-19 is obviously a very current and relevant topic. It would be interesting for the reader if this paragraph could be expanded somewhat and more explicitly discuss the reported association between eosinopenia and COVID-19 severity. On the other hand, steroids, which are of beneficial use in COVID-19 to decrease expression of virus receptors (Peters et al. 2020, Am. J. Respir. Crit. Care Med. 202:83-90) and decrease mortality, would be expected to decrease the number of eosinophils. Again, please expand on these complexities.
Author Response
Reviewer 2
This review manuscript describes neutrophil and eosinophil extracellular traps (NETs and EETs), including the mechanisms of DNA release and trap formation, as well as the function of traps in microbial defense and presence in disease. The manuscript also to some extent covers neutrophils and eosinophils more generally, and compares the two cell types. The subject is overall very timely. However, clarifications are needed at various places in the manuscript and expanded discussion on some items would be interesting for the reader.
We would like to thank the reviewer for the insightful comments that have significantly improved the manuscript. The detailed answer to each comment are provide below.
Major comments:
- Does vital NETosis include mitochondrial DNA and a similar mechanism as non-cytolytic EETosis?
Not all neutrophils die and release the mitochondrial DNA although the mere mitochondrial DNA can form traps (Yousefi). Now we added more information and references in this regard for further clarification.
- Do you believe that both viable and lytic EETosis happen in vivo or do you think vital mitochondrial DNA release may be a laboratory or experimental artefact? Do they both occur at the same time in the same diseases or are there differences?
EETosis happens in vivo and there are a lot of details of the clinical relevance of EETs in the new section dedicated to COVID. However, the knowledge about when they are form, if it is lytic or only mitochondrial DNA, and if those two mechanisms of EETosis happen simultaneously, are questions that still remain unanswer.
- E.g., on line 366: Please discuss how EETs can be targeted?
There is evidence that treatment with DNAse can improve lung pathology in some diseases, but there is now included on the manuscript a section on that discuss the use of anti-Siglec 8/f antibodies to improve artheriosclerosis and the results are very promising (Marx manuscript in Blood)
Minor comments:
- Line 98: Should it be “MPO” instead of “MOP” here?
Thanks for realizing the typo, not that has been corrected
- Line 140: What does the word “halt” mean and the entire sentence that includes the words “bacterial virulence factors are halt…”? Please rewrite.
We have changed this sentence “Within the NETs, bacteria cannot utilize the bacteria virulence factors to downregulate host immune response allowing for the immune system to better control the infection”
- Line 166: What is meant by “Neutrophils are able to recognize trigger NET formation…”? Recognize what?
We apology for this bad selection of words, that now has been changed by “Neutrophils are able to recognize different pathogens and trigger NET formation”
- Section 2.4 about NETs and inflammatory diseases including asthma: In this context, Lachowicz-Scroggins et al. 2019 (Am. J. Respir. Crit. Care Med. 199:1076-85) found NETs in sputum of a subset of patients with severe asthma and NETs were cytotoxic for airway epithelial cells.
Thanks for providing this reference that we have now included in section 2.4
- Line 233: Is “secondary granule” correct current nomenclature for eosinophils? Please consult recent reviews on eosinophil granules by Weller, Spencer and Melo.
We have now modified this based on the manuscript recommended “Eosinophils contain secondary granules eosinophil-specific granules130 or eosinophil secretory granules”
- Section 3.3 regarding eosinophils presented with immunoglobulins (Igs), etc.: Pathways involved in triggering cytolysis of and release of free granules from eosinophils activated by interleukin (IL)-3 and plated on aggregated IgG were recently described (Clin. Exp. Allergy 2020, 50:198-212).
We have now a new paragraph in section 3.5 dedicated to this manuscript. We consider that this manuscript will better fit in this section (3.5) as this section contains information about asthma.
- Lines 330-333: The relation between eosinophils and COVID-19 is obviously a very current and relevant topic. It would be interesting for the reader if this paragraph could be expanded somewhat and more explicitly discuss the reported association between eosinopenia and COVID-19 severity. On the other hand, steroids, which are of beneficial use in COVID-19 to decrease expression of virus receptors (Peters et al. 2020, Am. J. Respir. Crit. Care Med. 202:83-90) and decrease mortality, would be expected to decrease the number of eosinophils. Again, please expand on these complexities.
We would like to thank the reviewer for this suggestion. COVID-19 is a very interesting topic and we have dedicated a new section to NETs and COVID (2.3.1) and EETs and COVID (3.3.1)
Round 2
Reviewer 1 Report
Comments were adequately addressed. The manuscript is improved.
Author Response
We would like to thank the reviewer for the nice words.
Thanks
Reviewer 2 Report
General comments:
This revised review manuscript has essentially addressed all previous comments. However, with the new revised part of the text, a few new questions has arisen or a few clarifications are needed.
Minor comments:
- Line 85: Regarding bacterial modulation of eosinophils, you may want to add a literature reference here or does one or several references further down cover this area?
- Line 342: Is reference No. 130 really the correct one here?
- Lines 569-570: Please add a reference to the sentence starting with “In fact, in mice, SiglecF antibodies…”. Should it be ref. 214 also here as in the previous sentence or should it be another reference?
Author Response
We would like to thank the reviewer for the detailed comments we really appreciate the effort and the help.
- Line 85: Regarding bacterial modulation of eosinophils, you may want to add a literature reference here or does one or several references further down cover this area?
In this line, we were referring to a manuscript that was accepted at the time of the resubmission. We have now inserted the reference here as well as in the eosinophils section where it was necessary.
- Line 342: Is reference No. 130 really the correct one here?
Yes this was the reference were we got the term from
- Lines 569-570: Please add a reference to the sentence starting with “In fact, in mice, SiglecF antibodies…”. Should it be ref. 214 also here as in the previous sentence or should it be another reference?
We have now inserted the reference that is now 213 due to rearrangements
Overall, thanks a lot for the comments and suggestion that are improving the manuscript,